# Verification of the Four-Stage Model of Humor Processing: Evidence from an fMRI Study by Three-Element Verbal Jokes

**DOI:** 10.3390/brainsci13030417

**Published:** 2023-02-27

**Authors:** Chia-Yueh Chang, Yu-Chen Chan, Hsueh-Chih Chen

**Affiliations:** 1Department of Educational Psychology and Counseling, National Taiwan Normal University, Taipei 10610, Taiwan; 2Department of Educational Psychology and Counseling, National Tsing Hua University, Hsinchu 300043, Taiwan; 3Institute for Research Excellence in Learning Sciences, National Taiwan Normal University, Taipei 10610, Taiwan; 4Chinese Language and Technology Center, National Taiwan Normal University, Taipei 10610, Taiwan; 5NSTC AI Biomedical Research Center, Tainan 70101, Taiwan

**Keywords:** humor, middle frontal gyrus, inferior parietal lobule, amygdala, insula

## Abstract

The four-stage model comprises the expectation, incongruity, resolution, and elaboration stages of humor processing. In previous studies, most researchers used two-element jokes (setup and punch line) as stimuli, based on experimental methods, to explore the humor process. By contrast, the present study used a humor corpus with the novelty of three-element verbal jokes to perform direct separation from the material and clarify the humor processes. In this study, we used three-element verbal jokes and nonjokes, and we conducted a repeated-measures analysis of variance with a 3 × 2 two-way within-subject design. In humor processing, the posterior insula and middle frontal gyrus were mainly activated in the expectation; the middle temporal gyrus and the medial frontal gyrus in the incongruity; the inferior frontal gyri, superior frontal gyrus, and inferior parietal lobule in the resolution; and the ventromedial prefrontal cortex, amygdala, anterior insula, nucleus accumbens, and midbrain in the elaboration. The contributions of this study lie in its use of a humor corpus with the novelty of self-compiled three-element jokes, which not only successfully verified the models established in previous studies but added the expectation to the model; thus, this study separated the expectation and incongruity processes, making humor processing more complete.

## 1. Introduction

In recent years, neuronal activities observed during the humor comprehension and appreciation process have received attention in the field of cognitive neuroscience. An increasing number of studies are attempting to use neurophysiological measurements to explore the neural mechanisms of humor comprehension and appreciation in the brain [1,2,3,4,5]. According to humor processing studies that have examined both humor comprehension and humor appreciation, humor comprehension has been correlated with the activation of the frontal lobe cortex and temporal lobe cortex [6], whereas humor appreciation has been correlated with the activation of the mesolimbic system [7].

Scholars have attempted to clarify the process of humor comprehension and appreciation [4,6,8], as well as solely the process of humor comprehension [3,8]. Regarding the theory of humor processing, the incongruity–resolution theory [9] posits that an individual who perceives a joke to be humorous undergoes a process of incongruity before the resolution process. The comprehension–elaboration theory [10] suggests that humor comprehension consists of a process of incongruity and resolution, but that the process of incongruity and resolution only means that one has completed the comprehension of humor, which does not necessarily lead to a sense of amusement. After the incongruity and resolution process, it requires a process of elaboration in order to produce a sense of amusement. Chan [5] integrated the incongruity–resolution theory [9] and comprehension–elaboration theory [10] and designed a series of functional magnetic resonance imaging (fMRI) studies. The author performed in-depth investigations into the neural mechanism of humor processing, divided the process of an individual humor response into three stages, proposed the tri-component theory of humor (HTC), and constructed a three-stage neural circuit model for the process—from humor semantic comprehension to humor appreciation, and to the humor response [3,4,5]. The HTC [5], which is used in a model of humor processing called the cognition, affect, and laughter (CAL) model, refers to the following three-stage process: humor comprehension (cognition), humor appreciation (affect), and humor response (laughter).

In the HTC process, humor comprehension and humor appreciation are internal processes, while humor response is an explicit behavior. Humor comprehension is the cognitive process of humor, including the incongruity and resolution process, whereas humor appreciation refers to the state of pleasure that occurs following humor comprehension. In other words, while humor comprehension involves a process of incongruity and resolution, humor appreciation is a process of elaboration after humor comprehension, resulting in a sense of pleasure. Humor appreciation, therefore, adds an additional process (elaboration process) to humor comprehension. Lastly, the humor response refers to the person’s reaction to humor, namely the explicit behavior of laughter following the pleasant state. The present study focuses on the former part of the HTC process, which is the internal process of humor, with an emphasis on the comprehension and appreciation of humor. Thus, the three stages of humor processing are traditionally referred to as the incongruity, the resolution, and the elaboration stages.

According to the internal processes of HTC [5], the process of distinguishing humor comprehension and humor appreciation was previously divided into the three processes of incongruity, resolution, and elaboration. The first process of humor comprehension is “incongruity”, which is mainly the responsibility of the middle temporal gyrus (MTG) and the medial frontal gyrus (MFG). These regions of the brain are responsible for the detection of semantic conflicts and the storage of semantic information [3]. Chan and Chou et al. [3] compared participants’ brain activation when reading humorous, baseline (non-funny), and confusing sentences to separate the two stages of incongruity and resolution in humor comprehension. Their results indicated that the activation of the MTG and MFG was higher when the participants were reading confusing sentences than when they were reading baseline sentences; furthermore, these two brain regions played a role in the detection of incongruity and semantic conflicts, as well as the storage of semantic information in the process of humor comprehension. In addition, when participants read humorous sentences, the activation of the superior frontal gyrus (SFG) and inferior parietal lobule (IPL) was higher than that detected when they read confusing sentences; thus, these two brain regions were found to play a role in the process of humor comprehension.

The second process of humor comprehension is “resolution”, which mainly involves the inferior frontal gyri (IFG), SFG, and IPL. These brain regions dominate the functions of semantic comprehension and semantic integration. Chan and Chou et al. [4] compared participants’ brain activities when reading humorous sentences, baseline sentences (not incongruous and not funny), and “garden path sentences” (incongruous but not funny) to dissociate the components of cognition and affect in humor processing. When the participants read the garden path sentences, the activation of their IFG and SFG was higher than that observed when they read general sentences. These brain regions are associated with semantic comprehension, semantic selection, semantic integration, and other processes [3,11,12].

Finally, in the process of elaboration, the amygdala, midbrain, nucleus accumbens (NAc), ventromedial prefrontal cortex (vmPFC), parahippocampal gyri, and insula are highly activated when a “pleasant and funny feeling” is perceived [1,2,3,4,7,13,14,15,16,17,18,19]. For example, Chan and Chou [4] found that when participants read humorous sentences, the activation of their amygdala and vmPFC was higher than that when they read garden path sentences. In other words, these regions are highly correlated with incongruity resolution. The evaluation of funniness is strongly correlated with the vmPFC; in particular, an individual’s subjective sense of amusement is correlated with their vmPFC [6]. Humor appreciation and feelings of amusement are also correlated with the activation of the insula. In terms of the contrast between humor and nonhumor, the activation of the insula has been found to be higher when participants read humorous materials than when they read nonhumorous materials [17,18,19].

In previous studies of humor processing, scholars have mostly used jokes as stimuli. Traditionally, the structure of each joke is divided into two elements, namely, the setup and punch line. Setup involves the generation of the schema of the reader’s expectations; therefore, setup usually represents the process of expectation. The punch line includes the perplexity and resolution caused by the violation of expectations and the appreciative feelings and pleasant responses generated after comprehension; therefore, the punch line usually comprises three processes: incongruity, resolution, and elaboration [3,4,17,20]. In other words, in the correspondence between the humor structure of two-element jokes and the stages of humor processing, the first element is the setup, corresponding to expectation; the second element is the punch line, corresponding to incongruity (i.e., surprise), resolution, and elaboration (i.e., amusement).

The approach to exploring the process of humor processing mainly uses two-element jokes as the stimuli to conduct experiments [3,4,14,15]. Researchers may present setup sentences and the punch line on the screen simultaneously [7], or they may split the setup sentences and punch line into two steps that appear sequentially on the screen [3,4,8]. However, as the punch line contains too many humor processes, it is not possible to observe them directly, and thus experimental manipulation is required to dissociate the different processes. In other words, previous studies required the use of different types of jokes to explore the process by comparing the contrasts of the punch line. Taking the two-element joke in Table 1 as an example, in the punch line, the “humor sentence” contains incongruity, resolution, and elaboration processes. The “garden path sentences” contain incongruity and elaboration processes. The “baseline sentence (nonjoke)” is without incongruity, resolution, and elaboration processes. By comparing the “humor sentence” with the “garden path sentences”, we could obtain the process of elaboration, and by comparing the “garden path sentences” with the “baseline sentence (nonjoke)”, we could obtain the process of incongruity and resolution [4].

In other words, most of the previous studies have adopted an experimental design approach by manipulating punch line sentences to dissociate the process of incongruity, resolution, and elaboration contained within the punch line. However, it could scarcely be dissociated from the setup, as the setup sentences of jokes are usually the same, and therefore can scarcely be dissociated from the activation of the individual’s brain area during the experience of expectation when engaging in a humorous experience. In the traditional method of contrast comparison, there is inevitably inaccuracy because the comparison by contrast within the punch line requires the assumption of a linear model.

This study identifies a type of joke whose structure has three elements, named the “three-element joke”. The three-element jokes allow for a more refined cut-up of the process contained in the punch line, splitting it into two elements: a first punch line and a second punch line. Taking Table 2, for example, the first element is the setup, which corresponds to expectation, and the punch line has two sentences—the first is the first punch line, which corresponds to incongruity, and the second is the second punch line, which corresponds to resolution and elaboration. The advantage of the three-element joke is that they enable the comparison of contrasts within a single joke—that is, they can directly distinguish between different processes in terms of stimuli, thus reducing inaccuracy. In this way, the humor structure can directly correspond to humor processing, and the four-stage model of humor processing can be validated more effectively. In the present study, the four-stage model of humor processing is named the four-stage model of humor processing and comprises the expectation, incongruity, resolution, and elaboration stages of humor processing.

When studying humor issues, the humor materials used are mainly verbal and nonverbal materials, and the verbal materials are mostly jokes. Because verbal materials can have exquisite and complex meanings and offer advantages when studying the process of humor processing, many studies have used jokes as research materials [3,4,5,8,9,20,21]. Nonverbal images are mainly funny pictures, and the reflected content includes both text and mental images; therefore, they are commonly used by researchers [7,9,12,14,15,16,22,23].

However, from the perspectives of humor structure, as well as the processes of humor comprehension and appreciation, few studies have specifically focused on the front end of the humor structure, known as motivation (i.e., the expectation stage). Chan and Hsu [14] and Chan and Hsu [15] used the reward of humor as a theme, designed experiments with anticipation (i.e., expectation) and outcome phases, and presented humor or money as the material for distinguishing the two processes. Anticipation represents the motivational brain state, indicating the extent to which the participant desires the subsequent reward. The outcome represents the hedonic brain state, indicating the response of the participant after receiving different rewards. The authors’ results revealed that the NAc is activated when the reward is a general monetary reward, the amygdala is mainly activated in response to a humor reward, and the posterior orbitofrontal cortex (pOFC) is mainly activated for a sexual humor reward.

The present study differed from previous studies on humor in the approach that it used to divide humor processing in an attempt to clarify the various stages of humor. Previously, methods for dividing humor stages have been based on the experimental design and using two-element jokes, with the process being divided into the three stages of incongruity, resolution, and elaboration [3,4]. The present study used the design of humor materials, directly divided the processes using the structure of humor texts, the “three-element jokes”, and attempted to use the three elements of setup, first punch line, and second punch line to better clarify and fit the humor process of expectation, incongruity, resolution, and elaboration.

Based on previous research on humor processing [3,4], an analysis of regions of interest (ROIs) was performed in this study [24]. The ROIs of different stages are listed as follows: the two ROIs of the incongruity stage are the MTG and MFG; the three ROIs of the resolution stage are the IFG, SFG, and IPL; and the five ROIs of humor and pleasure are the vmPFC, amygdala, insula, NAc, and midbrain. 

The purpose of this study was to use innovative materials to verify the existing process of humor processing. The innovation of this study is that we self-compiled jokes with three elements (setup, first punch line, second punch line). The three-element jokes can not only map setup to expectation but also refine the process contained in the punch line by splitting the first punch line (to incongruity) and the second punch line into resolution and elaboration. This is the innovation that characterizes our study, as the three-element jokes that we used allowed for the more direct validation of the four-stage model, as it corresponds more appropriately to humor processing (expectation, incongruity, resolution, and elaboration).

This study developed and tested the following hypotheses. First, this study predicted that the effect of the materials used would be the same as that of the materials used in previous humor studies, and also that the materials used in this study might enable people to feel pleasant. Second, this study assumed that the three-element verbal jokes used might also successfully separate humor processing, as in previous experimental designs. Finally, the research material used in this study had the humor elements of setup, first punch line, and second punch line and was based on previous humor theories; therefore, this study expected to expand the existing humor processing framework to a four-stage model, adding the expectation stage at the forefront of the process to make the process of humor processing more complete.

## 2. Materials and Methods

### 2.1. Participants

A total of 54 healthy volunteers (27 men and 27 women; mean age = 24.02 years, *SD* = 4.41, range = 20–40 years) participated in this study. They were all right-handed native speakers of Mandarin [25] and had no history of mental or neurological disorders. This study was approved by the Research Ethics Committee of the National Taiwan University. All participants provided written informed consent before the study commenced.

### 2.2. Stimuli

A joke is usually divided into a humor comprehension process and a humor appreciation process. The present study employed verbal jokes instead of cartoons. Joke structure is divided into the three elements of setup sentences (stage 1, expectation), first punch line sentences (stage 2, incongruity), and the second punch line (stages 3 and 4, resolution and elaboration) [3,4]. This study drew on humorous materials from the Internet, books, magazines, and other researchers [26]. A total of 90 three-stage verbal jokes were selected by four experts, and each stage was rated for funniness, unusualness, and comprehensibility to ensure their representativeness. Then, 60 jokes were selected that had setup sentences with a low level of funniness, low level of unusualness, and high level of comprehensibility; a first punch line with low levels of funniness, high levels of unusualness, and low levels of comprehensibility; and a second punch line with high levels of funniness, low levels of unusualness, and comprehensibility.

The joke and nonjoke stimuli used in this study’s experiment were paired (Table 3). During the first punch line and second punch line, for both funniness and nonfunniness, the nonjoke material was rewritten from the paired joke material—that is, the joke’s first punch line sentence and second punch line sentence were rewritten as straightforward sentences and non-incongruity, which were the baseline conditions for the experiment, with a total of 60 nonjokes. The average word count of the joke and nonjoke setup sentences was 36.72 (*SD* = 2.23, range = 34–41); that of the joke first punch line sentences was 27.16 (*SD* = 2.64, range = 22–32); that of the joke second punch line was 20.07 (*SD* = 2.61, range = 16–25); that of the nonjoke first punch line sentences was 27.38 (*SD* = 1.78, range = 22–33); and that of the nonjoke second punch line was 20.38 (*SD* = 1.42, range = 18–25). We have provided several additional three-element jokes in the Appendix A, with both original Mandarin Chinese and English translations attached, for reference.

### 2.3. Experimental Paradigm

This experiment was based on a 3 × 2 two-way within-subject design, and an fMRI analysis was performed. The first independent variable was the sentence “stage,” which included “expectation,” “incongruity,” and “resolution–elaboration”; the second independent variable was the “type,” which included “joke” and “nonjoke.” Thus, there were a total of six conditions; namely, joke expectation (EJ), joke incongruity (IJ), joke resolution–elaboration (RJ), nonjoke expectation (EJ), nonjoke incongruity (IJ), and nonjoke resolution–elaboration (RJ). The in-scan behavioral-dependent variable was the funniness of the humor corpus as rated by participants, and the dependent variable of brain activity was the change in the blood oxygenation level dependent (BOLD) after the participants’ response, such as humor comprehension and humor appreciation.

This study consisted of 4 runs, each with 15 trials, for a total of 60 trials. The intervals were presented in a counterbalance order, and the trials within the interval were presented in a random manner. This study comprised the following three procedures: instructions and practice before undergoing magnetic resonance imaging (MRI) scanning; formal research after entering the MRI scanner; and debriefing after the completion of the post-scan.

Before the formal scan, the researcher explained the relevant information of the experiment to the participants, including its purpose, the procedure, the participants’ rights and interests, and the safety of MRI scanning; furthermore, the participants signed informed consent forms. Next, a six-question practice trial was conducted, in which the characteristics of the material and the presentation procedure were the same as in the formal experiment. Participants entered the fMRI scanning room for the formal experiment only after they fully understood and were familiar with the procedure. Before officially entering the scanner, the participants were checked again to ensure that no metal objects were present on their bodies. Subsequently, under the guidance of the operator, the participants laid on their backs in the magnetic field, and they were asked to keep their heads as still as possible and to breathe as smoothly as possible. Before the formal experiment, a structural scan (T1 anatomical image) was performed. During the formal experiment, a functional scan (e.g., echo-planar imaging [EPI]) was performed.

Four runs were completed in the formal experiment. At the beginning of each run, a dummy scan was performed first, and the participants were then asked to focus on a fixation point in the center of the screen for 500 ms. The first humorous sentence was presented for 7320 ms, the second was presented for 5760 ms, and the third was presented for 6750 ms. A random jittered interstimulus interval (ISI) of 700–1200 ms was presented in the sentence, with the average being 1 s. Then, a funniness scale of 1–4 was presented. Each rating was presented for 3.5 s, and the participant used the index, middle, ring, or little finger of their right hand to indicate a rating of one, two, three, or four points, respectively. A random jittered intertrial interval (ITI) of 700–1200 ms was presented in each trial, with the average being 1 s. A 1 min rest period with eyes closed was implemented at the end of each run. Figure 1 presents a procedure of the experiment.

### 2.4. Image Acquisition

In this study, fMRI scans were performed using a 3T Siemens Magnetom Prisma scanner (Erlangen, Germany) with a 20-channel head coil with a 90° phase difference and input/output functions. The experiment was performed at the Imaging Center for Integrated Body, Mind and Culture Research at the National Taiwan University. The experimental program was written using E-prime 3.0, and the current design 1 to 4 keyboard of the center was used. The stimuli were visualized on goggles worn by the participants. The image acquisition parameters were as follows: a repetition time (TR) of 2000 ms, an echo time (TE) of 26 ms, and a flip angle (FA) of 90°. The entire head (whole brain) was scanned using the axial view aligned with the anterior commissure–posterior commissure (AC-PC line), and 1 TR scan of the whole brain comprised 40 slices. The field of view (FOV) was 220 × 220 cm^2^, the matrix size was 64 × 64, the voxel size was 3.4 × 3.4 × 3.4 mm, and the thickness of each slice was 3.4 mm. The dummy scan was 3 TR (i.e., the scan of the first 6 s of each interval was not included) and a maximum of 210 TRs were scanned during each run.

### 2.5. Image Analysis

The imaging data analysis procedure was divided into the two stages of preprocessing and statistical analysis. In this study, the Statistical Parametric Mapping 12 (SPM12; Wellcome Department of Cognitive Neurology, London, UK) software package was used to process and analyze the fMRI images. After the T1 and EPI images scanned at the fMRI center were first imported using the Digital Imaging and Communication in Medicine (DICOM) standard, preprocessing was performed. The preprocessing steps were slice timing, realignment, coregistration, normalization, and smoothing.

First, in the slice timing step, because most of the pulse sequences were used to acquire data in a cross-sectional manner, the scanner first acquired the odd-numbered sections and then the even-numbered sections to reduce cross-sectional excitation or activation. Therefore, the interpolation method was used to correct the timing of interleaved sections and minimize the time error. The alignment reference point was the midpoint of a total of 40 slices (i.e., the 40thslice was used as the reference point).

Second, in the realignment step, the participants’ heads were moved and realigned to ensure that brain artifacts captured in the wrong position were not analyzed. According to the combination of three translations (moving the entire volume along the *x*, *y*, and *z* axes) and three rotations (rotating the entire image volume through the x-y, y-z, and x-z planes), the participant’s head movements were within a reasonable range during each run, if the head movement was violent and irregular, the data were deleted and no further analysis was performed.

Third, in the coregistration step, alignment and calibration were performed for T1 and EPI scans in sequence, with the mean image selected in the reference image and T1 selected in the source image. Fourth, in the normalization step, mathematical expansion, compression, and deformation were used to map each brain to the standard brain. In this study, the standard brain created by the Montreal Neurological Institute (MNI) was used as the standardized EPI template, and the voxel size was adjusted to 3 × 3 × 3.

Lastly, the purpose of the smoothing step was to increase the signal-to-noise ratio (SNR). Since noise usually has a relatively high spatial frequency, the fuzzy technique of the Gaussian kernel filter is commonly used to normalize the distribution of errors. The width of a typical filter is set to 6–10 mm full width at half-maximum (FWHM; i.e., approximately two or three voxels). In this study, the Gaussian kernel 6 mm FWHM filtering method was used.

In the analysis of functional brain imaging data, the statistical analysis comprised two stages, namely a first-level individual analysis and a second-level group analysis. In terms of the individual analysis, this study used canonical hemodynamic response functions (HRF) to estimate the brain mechanisms of each participant when they were reading the three-element jokes and nonjokes. The general linear model (GLM) was used for the statistical analysis, and the six head movement parameters were used as covariates to eliminate the errors caused by head movement. In this study, a whole-brain voxel-based functional activation analysis was performed; the threshold was a peak-level familywise error (FWE)-corrected *p* < 0.05; small volume correction (SVC) was also performed; and the cluster voxels were greater than 10.

## 3. Results

### 3.1. In-Scan Behavioral Results

Participants’ funniness ratings were scored on a four-point Likert scale during scanning. The mean rating scores were 2.76 (*SD* = 0.60) for the joke condition and 1.46 (*SD* = 0.40) for the nonjoke condition. A paired-sample *t*-test of participants’ rating scores revealed a significant difference, where *t* (53) = 19.04, *p* < 0.001, and Cohen’s *d* = 2.55.

#### 3.2. fMRI Results

This study focused on a comparison between the three stages of jokes (expectation, incongruity, and resolution–elaboration; corresponding to three-element jokes, with setup, first punch line, and second punch line); thus, an analysis of the interaction between stage and type (3 × 2) was conducted first, followed by an analysis of the simple main effects of stage and type.

##### 3.2.1. Interaction of Stages and Types (Interaction Analysis)

Table 4 presents the results of the interaction analysis. An interaction between stage and type was observed in the right MTG, bilateral MFG, bilateral IFG, bilateral SFG, right IPL, bilateral vmPFC, bilateral amygdala, bilateral midbrain, and right insula. The simple main effect of the type and the stage is presented in Table 5 and Table 6, respectively.

##### 3.2.2. Simple Main Effect of Each Type

The greatest difference between this study and previous studies is that this study used the material to separate the humor comprehension and appreciation processes. Therefore, based on previous studies on humor, this study mainly focused on the differences between jokes and nonjokes (baseline) in the resolution and incongruity stages. The nonjoke conditions (EN, IN, and RN) were the baseline conditions.

Table 5 summarizes the simple main effects of the joke type. Since the setup sentences in the jokes and nonjokes were the same, the contrast between the joke and nonjoke (EJ > EN) exhibited no significant differences in the *expectation stage*. In the *incongruity stage*, the contrast between the joke and nonjoke (IJ > IN) revealed greater activation in the left IFG. In the *resolution and elaboration stage*, the contrast between the joke and nonjoke (RJ > RN) revealed greater activation in the right MTG, bilateral MFG, left MiddleFG, bilateral IFG, right SFG, right IPL, left vmPFC, bilateral amygdala, left midbrain, and left NAc.

##### 3.2.3. Simple Main Effect of Each Stage

Table 6 summarizes the simple main effects of the stage. This study focused on a comparison between different elements of joke materials, aiming to use the research material in a manner that separated the humor comprehension and humor appreciation processes.

The comparison between the expectation and incongruity stages (EJ > IJ) revealed greater activation in the bilateral MiddleFG, bilateral IFG, right SFG, and left vmPFC. Furthermore, the comparison between the setup and resolution–elaboration stages (EJ > RJ) revealed greater activation in the right MFG, bilateral MiddleFG, left IFG, right SFG, right vmPFC, and bilateral insula. Figure 2 presents stage 1: setup.

Moreover, the comparison between the incongruity and expectation stages (IJ > EJ) indicated greater activation in the bilateral MTG, bilateral MFG, bilateral MiddleFG, bilateral IFG, bilateral SFG, right IPL, bilateral midbrain, and bilateral insula. In addition, the comparison between the incongruity and resolution–elaboration stages (IJ > RJ) revealed greater activation in the right IFG, right SFG, and bilateral insula. Figure 3 presents stage 2: incongruity.

Furthermore, the comparison between the resolution–elaboration and expectation stages (RJ > EJ) revealed greater activation in the bilateral MTG, left MFG, left IFG, bilateral SFG, left vmPFC, bilateral amygdala, bilateral midbrain, bilateral NAc, and bilateral insula. Moreover, the comparison between the resolution–elaboration and incongruity stages (RJ > IJ) revealed greater activation in the right MTG, left MFG, bilateral MiddleFG, bilateral IFG, bilateral SFG, bilateral IPL, bilateral vmPFC, bilateral amygdala, bilateral midbrain, bilateral caudate head (close to the NAc), and bilateral insula. Figure 4 presents stage 3: resolution. Figure 5 presents stage 4: elaboration.

## 4. Discussion

The results of this study are consistent with those of previous studies. In humor processing, the MTG and MFG are activated in the incongruity stage; the IFG, SFG, and IPL are mainly activated in the resolution stage; and the vmPFC, amygdala, anterior insula (AI), NAc, and midbrain are mainly activated in the elaboration stage. In addition, a finding that differed from those of previous studies was noted in the expectation stage. We analyzed this stage and found that the main regions activated were the MFG and posterior insula (PI).

Regarding the methodological, theoretical, and practical contributions of this study, firstly, the most valuable methodological contribution of the present study is the innovation of the humor material. The three-element jokes self-compiled by the study can effectively segregate the processes contained in the punch line. The difference between the present study and previous studies is that previous studies used two-element jokes (including setup and punch line), but the punch line contains several humor processing stages (incongruity, resolution, and elaboration). The three-element jokes (setup, first punch line, and second punch line) used in the present study allow for a more accurate comparison of the humor structure with humor processing, which could more accurately validate the four-stage model of humor processing in the neuroscience-based framework.

Secondly, the theoretical contribution of this study is to propose the four-stage model of humor processing (expectation, incongruity, resolution, and elaboration), which expands the process of expectation from the traditional three-stage model (incongruity, resolution, and elaboration). In previous studies on humor comprehension and appreciation processing, the setup (expectation stage) was rarely discussed in particular. In the present study, three-element jokes could be used to effectively segment the process of expectation. In addition, three-element jokes are also beneficial for future ongoing humor research.

The practical contribution of this study is mainly based on the benefit of humor training. Traditionally, the punch line is a combination of many stages at the same time, which makes it difficult to teach humor training. The three-element humor materials compiled by this study will help to provide corresponding materials in different stages of humor training in the future, which will help individuals to understand and appreciate humor and develop a sense of humor. In terms of humor production, the three-stage humor process identified in this study can also be used to provide examples of humor production for individuals, which can be helpful in humor training and thus help to improve one’s physical and mental health.

This study used three-element jokes—that is, the material itself was used to identify and distinguish the four-stage process of the humor processing of verbal jokes in terms of neurophysiology. Previous studies [3,4,12] have generally used experimental designs to dissociate humor processing by performing comparisons between materials. Most have attempted to separate humor processing into the two elements of setup and punch line. The innovation of this study lies in its use of three-element jokes. In other words, the joke itself covered the three processes of expectation, incongruity, and resolution–elaboration, and, thus, humor processing was dissociated using one material. This approach reduces errors in comparisons between materials and is also rigorous in terms of material innovation.

The first objective of this study was to explore the effectiveness of the materials and to verify the first hypothesis, namely that the resolution–elaboration stage of the joke (RJ > RN) affects humor appreciation. The results indicated significant activation in the amygdala, vmPFC, midbrain, VTA, and NAc, and these brain regions all support humor elaboration. In addition, in the incongruity stage (IJ > IN), activation of the IFG was observed. Noteworthily, previous studies have demonstrated that the IFG is the region activated during incongruity and resolution, whereas the present study found that the IFG was activated during the incongruity stage.

The second objective of this study was twofold: to verify that the three-element jokes used in the analysis successfully separated the four stages of humor processing, and to use the contrast between incongruity and expectation and the contrast between resolution–elaboration and incongruity to separate humor processing, which has mostly been separated by the punch line in previous studies. In the traditional three-stage process of humor processing, the more active brain regions, as revealed by the comparison between incongruity and expectation (IJ > EJ), were first used to represent the process of incongruity. The results were consistent with those of previous studies, confirming greater activation of the MTG and MFG; furthermore, these two brain regions were more strongly correlated with cognitive and executive functions in the process of humor comprehension, as well as being involved in the processing of language content fluency. The MTG is more relevant to semantic comprehension in the humor comprehension process [6]; thus, it dominates the detection of humor, semantic information storage, semantic comprehension, and integration [3,4]. The activation of the MFG is correlated with the use of cognitive resources for detecting humor stimuli [27]. Moreover, a study reported that the MFG is also correlated with the examination, decoding, and comprehension of humor stimuli [28].

In addition, as a representative analysis of the resolution process, a comparison between resolution and incongruity (RJ > IJ) was performed to reveal the most active regions. The results revealed that the IFG, SFG, and IPL exhibited significant activation. The process of incongruity resolution should involve schema conversion, ideological integration, and information collation to remove ambiguity, while the IFG, SFG, and IPL all involve the processing of executive functions and are related to them. The IFG involves the resolution of semantic ambiguities and schema conversion in verbal jokes [3,4,6,27,29]. In particular, the activation of the left IFG involves both the verbal comprehension processes of disambiguation and resolution in verbal sentences [27] and the search for and integration of past experience [30]. The SFG participates in high-level cognitive processing and previous studies have found it to be mostly activated in the context of incongruity [3,4,27,30]. During humor comprehension, individuals must organize and integrate internal contextual consistency, which is correlated with SFG activation [30]. Samson and Hempelmann [30] reported that the activation of the IPL may be correlated with the comprehension of the causal relationship between the setup and the punch line, which is consistent with the results reported by Chan and Chou [3] as well as Dai and Chen [29]. Greater activation of the IPL is observed in the stages of incongruity and resolution, potentially because the IPL is involved, where individuals generate a new or consistent interpretation of an unexpected situation to understand the relationship between the expectation, incongruity, and resolution–elaboration [3].

Finally, as a representative analysis of the process of humor elaboration, a comparison between resolution–elaboration and incongruity (RJ > IJ) was conducted to reveal more active regions, which was similar to the resolution process. The difference is that the former focuses on cognitive activities, whereas this process focuses on emotion-related brain regions. The amygdala, vmPFC, midbrain, SN, and AI were all highly activated. In the present study, high NAc activation was observed in the comparison between resolution and expectation (RJ > EJ). The amygdala is a critical structure related to reward and involves salience processing [1]. In recent years, many studies have supported the critical status of the amygdala in humor appreciation [1,2,3,4,7,13,14,15]. Vrticka and Black [1] reviewed the function of the amygdala in the process of humor appreciation, noting that humor appreciation may activate the amygdala since this region may be related to incongruity detection and problem solving and possibly to the social process—that is, the amygdala is involved in individuals’ humor comprehension and humor appreciation and is closely related to the humor reward mechanism. In addition, the amygdala may be involved in the process of focusing on humorous stimuli and possess the function of assessing emotions when detecting incoming stimuli [13]. Therefore, consistent with our expectations and previous studies, we observed the activation of the bilateral amygdala during the stage of humor elaboration. In addition, in said stage, other brain regions related to emotions were found to be activated, which is consistent with previous studies. The vmPFC is strongly correlated with the funniness of humor subjectively perceived by individuals [6], and its activation is higher in the stage of humor elaboration [4]. The regions related to sensory pleasure also include the midbrain, SN, AI, and NAc, which we verified to be crucial in the pleasurable response to humor appreciation. These regions are associated with the subjective awareness of funny feelings and the experience of positive reward feelings, and they display higher activation in the humor appreciation process after the resolution stage [4,27,29].

Notably, few studies have explored the expectation stage. The advantage of this study was that the three-element materials could be used to directly compare the stages. Therefore, based on the established three-stage process of humor processing [3,4,5] and the findings described above, the third main objective of this study was to explore the expectation process by comparing the contrast between expectation and incongruity (EJ > IJ), as well as that between expectation and resolution (EJ > RJ); then, we aimed to add another process, namely “expectation”, to extend the established three-stage process of humor processing. We compared the contrast between expectation and incongruity (EJ > IJ) to reveal regions with higher activation, revealing the process of the exclusion of incongruity; therefore, higher activation was observed in brain regions targeting the expectation process, particularly the right MiddleFG. Moreover, the contrast between expectation and resolution (EJ > RJ) was compared to reveal regions with higher activation, revealing the process of exclusion of resolution; therefore, higher activation was observed in brain regions targeting the expectation process, particularly the PI. Thus, during the expectation stage, the brain generated schema for the detected information and began attempting to understand the meaning of the text, since the activation of the MFG was related to the inspection, decoding, and comprehension of humor stimuli [28]. The activation of the PI in the expectation stage was consistent with the latest result published by Chan [31], namely that the PI is related to the somatosensory information generated by individuals. The author compared the setup and punch line and demonstrated higher activation of the PI in the setup; however, higher activation of the AI was observed in the punch line, which is also consistent with the present study. In previous studies examining the motivation of humor (i.e., the anticipation stage), Chan and Hsu [14,15] used pictures as the humor material and did not include the insula as an ROI. Their results indicated that the main activated region was the NAc. In the present study, the insula was used as an ROI and verbal jokes were presented as the humorous material. Possible explanations for this discrepancy with previous studies are as follows: (1) the materials used in this study differed from those used in previous studies; and (2) the insula was not used as an ROI in previous studies. We will explore these factors in future studies.

In addition, this study differed from the previous divisions of the three stages of incongruity, resolution, and pleasure [3,4]. This study used expectation, incongruity, resolution, and elaboration, which may be more consistent with the structure of humor in the process of humor comprehension and appreciation. Our results not only support but also expand the three-stage brain neurological process of humor processing proposed in previous studies to a four-stage model, and we also classified the process of humor processing into four stages, namely expectation, incongruity, resolution, and elaboration, as depicted in Figure 6. In this model, the MiddleFG and PI may be involved in expectation (i.e., the expectation and perception of the content during the initial reading). Subsequently, the MTG and MFG may be involved in the process of incongruity, and the IFG, SFG, and IPL may be involved in the process of resolution, which includes the integration of semantic content. Finally, brain regions such as the vmPFC, amygdala, AI, NAc, and midbrain may be involved in the process of elaboration, resulting in a sense of pleasure and amusement. The results of this study are consistent with those of previous studies [3,4,5], while the present study also further expanded the process of humor processing by adding one initial stage, making the overall neurological process of humor comprehension and appreciation more complete.

## 5. Conclusions

The present study adopted innovative techniques and used three-element verbal jokes to successfully separate the process of humor comprehension and appreciation into four stages, which comprised expectation, incongruity, resolution, and elaboration. Expectation may involve the activation of the MiddleFG and PI [31]; incongruity may be related to the activation of the MTG and MFG [3]; resolution may be related to the activation of the IFG, SFG, and IPL [3]; and elaboration may be related to the activation of the vmPFC, amygdala, AI, NAc, and midbrain [3,4,27,29].

Most previous studies have used two-element (setup and punch line) jokes as stimuli and have been based on an experimental design and cross-material methods, such as comparisons of funny jokes with nonfunny nonjokes, as well as comparisons of material with incongruity and resolution with material that has incongruity but no resolution (i.e., a comparison of joke types between sentences) [3,4]. Furthermore, previous studies have mainly conducted comparisons in the two elements of setup and punch line to clarify the process of humor processing. The main contribution of this study lies in its use of a specific three-element joke. The jokes themselves had a three-element structure, which comprised a joke setup sentence (expectation), a first punch line sentence (incongruity), and a second punch line sentence (resolution and elaboration); therefore, this study was able to perform comparisons within sentences. Through the comparison of resolution and incongruity, as well as of incongruity and expectation, the process of humor processing could be clarified within the same material in more detail. The advantages of using this approach are that the process of humor comprehension can be clarified in more detail and the errors between materials can be reduced. Through a comparison between the three elements of jokes, this study verified and expanded the existing humor process.

## Figures and Tables

**Figure 1 brainsci-13-00417-f001:**
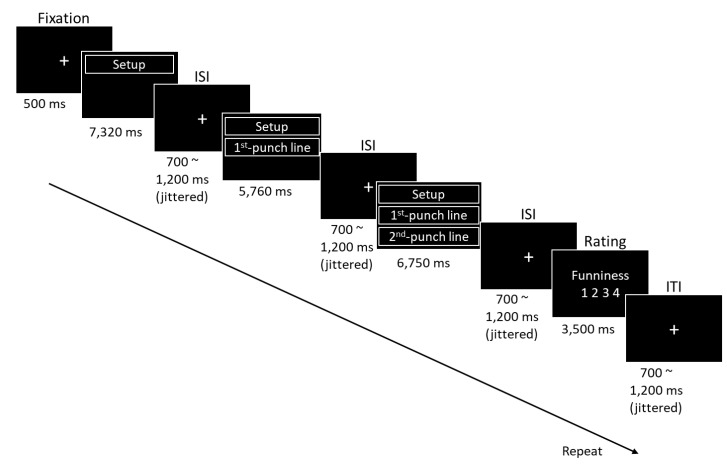
Experimental procedure.

**Figure 2 brainsci-13-00417-f002:**
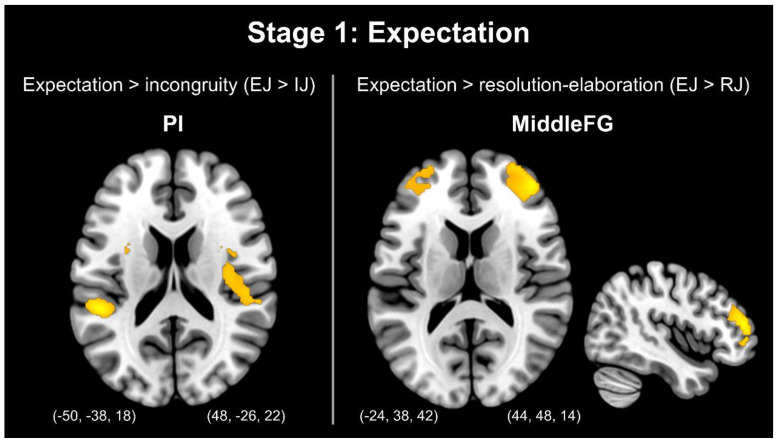
Stage 1: Expectation. Note: PI = posterior insula; MiddleFG = middle frontal gyrus.

**Figure 3 brainsci-13-00417-f003:**
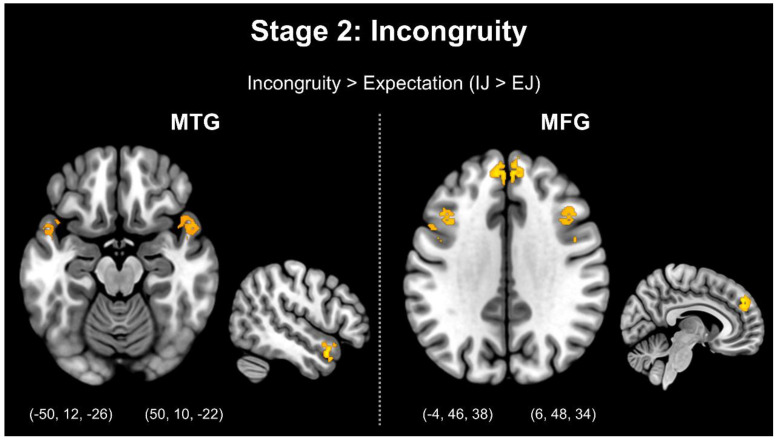
Stage 2: Incongruity. Note: MTG = middle temporal gyrus; MFG = medial frontal gyrus.

**Figure 4 brainsci-13-00417-f004:**
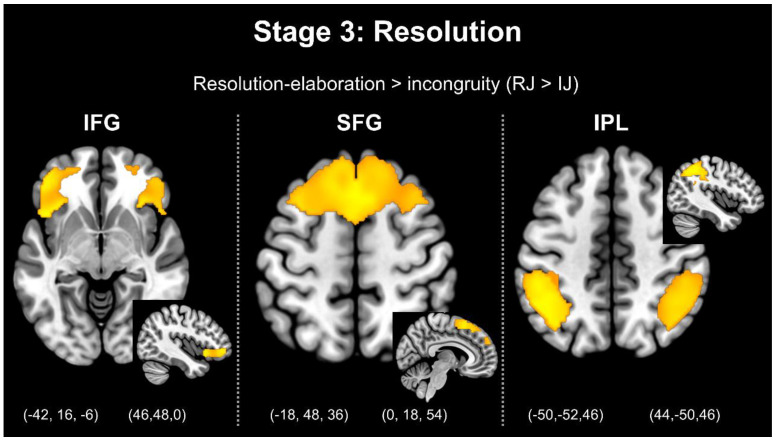
Stage 3: Resolution. Note: IFG = inferior frontal gyrus; SFG = superior frontal gyrus; IPL = inferior parietal lobule.

**Figure 5 brainsci-13-00417-f005:**
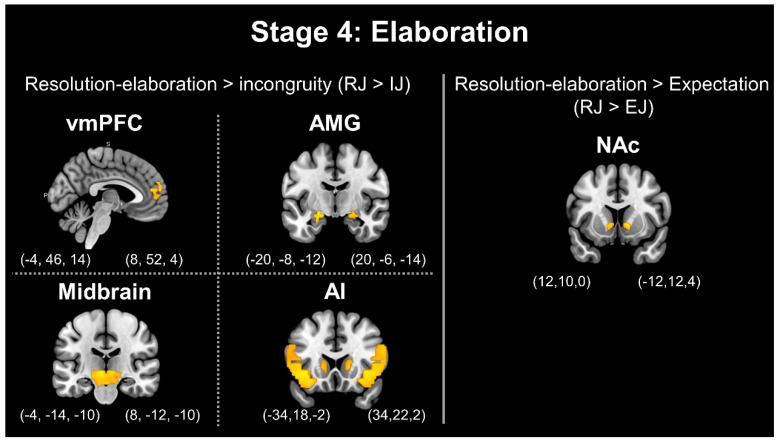
Stage 4: Elaboration. Note: vmPFC = ventromedial prefrontal gyrus; AMG = amygdala; AI = anterior insula; NAc = nucleus accumbens.

**Figure 6 brainsci-13-00417-f006:**
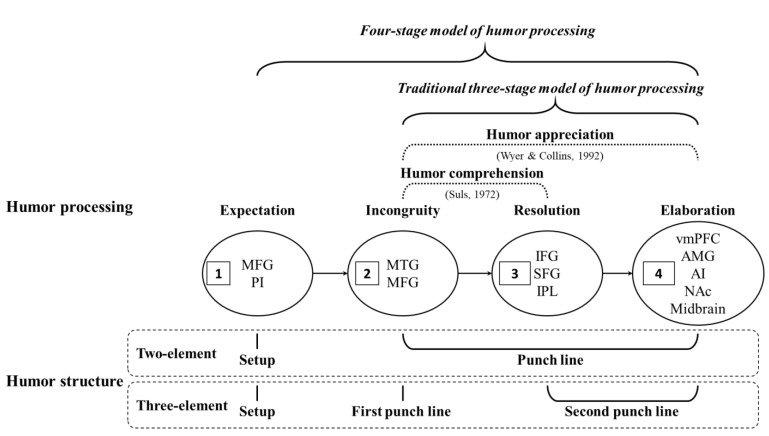
The four-stage model of humor processing. Humor processing and humor structure correspondence diagram. Four stages of the neural circuit underlying humor processing: expectation, incongruity, resolution, and elaboration. Figure 6 was expanded by the three-stage brain neurological process of humor comprehension and appreciation [3]. Note: MFG (in expectation stage) = middle frontal gyrus; PI = posterior insula; MTG = middle temporal gyrus; MFG (in incongruity stage) = medial frontal gyrus; IFG = inferior frontal gyrus; SFG = superior frontal gyrus; IPL = inferior parietal lobule; vmPFC = ventromedial prefrontal gyrus; AMG = amygdala; AI = anterior insula; NAc = nucleus accumbens.

**Table 1 brainsci-13-00417-t001:** An example of a two-element joke.

Humor Structure (Humor Processing)		
Setup (Expectation)	Punch Line (Incongruity, Resolution, and Elaboration)	
One day after work, a mother buys some donuts from a store close to her office. When she gets home, she says to her eldest son, “Peter, Mom brought some sweets home. You can take one donut to share with your little brother. Don’t eat it all yourself!” So Peter takes the donut, thanks his mom, goes to his little brother and says,	humor sentence	“Hey, we have a donut to share! I’ll take the circle, and you can have the hole!”
	garden path sentence	“I don’t eat chocolate donuts are more to my taste.”
	baseline sentence (nonjoke)	“Hey, we have a donut to share! I’ll eat half and you can have the other half!”

**Table 2 brainsci-13-00417-t002:** An example of a three-element joke.

Humor Structure (Humor Processing)		
Setup (Expectation)	First Punch Line (Incongruity)	Second Punch Line (Resolution and Elaboration)
The director of the psychiatric hospital: “Tomorrow the president will come, you all have to clap your hands, and if you do well, you will get a meat bun.” The next day, the patients did clap wildly.	Suddenly, a patient rushed out, reached out and slapped the president in the face.	“You don’t want to eat meat buns?” The patient said fiercely.

**Table 3 brainsci-13-00417-t003:** An example of a three-element joke and nonjoke.

	Humor Structure (Humor Processing)		
	Setup (Expectation)	First Punch Line (Incongruity)	Second Punch Line (Resolution and Elaboration)
Joke	The director of the psychiatric hospital: “Tomorrow the president will come, you all have to clap your hands, and if you do well, you will get a meat bun.” The next day, the patients did clap wildly.	Suddenly, a patient rushed out, reached out and slapped the president in the face.	“You don’t want to eat meat buns?” The patient said fiercely.
Nonjoke		At the end of the day, the director prepared the meat buns according to the quantity promised, and everyone got a big meat bun.	They all enjoyed the food and agreed that it was a good job.

**Table 4 brainsci-13-00417-t004:** Brain regions associated with interactions of the expectation, incongruity, and resolution-elaboration stages with joke types.

				MNI Coordinates			
Brain Region	BA	Voxels	Side	*x*	*y*	*z*	*F* Value
(1) Interaction of Stages and Joke Types							
MTG	38	29	R	50	10	−22	18.49
MiddleFG	8	262	R	32	24	46	16.30
	9	14	L	−32	42	32	12.48 ^‡^
IFG	10	32	L	−42	40	14	18.69
	-	23	R	44	46	4	11.71 ^‡^
SFG	6	136	R	10	6	66	14.85
	10	25	L	−20	56	26	8.57 ^‡^
IPL	-	1094	R	60	−44	40	26.87
vmPFC (MFG)	11	247	L	−4	32	−16	20.31
	9	11	R	6	52	18	8.40 ^‡^
Amygdala	-	918	L	−20	−8	−14	69.64
		407	R	22	−6	−16	41.83
Midbrain	-	474	L	−14	−10	−12	29.91
Midbrain (VTA)		132	L	−4	−16	−8	20.19
Midbrain (SN)		9 ^†^	L	−6	−14	−10	14.51 ^‡^
		11	R	10	−20	−12	11.36 ^‡^

Notes: The activation threshold was set to *p* < 0.05 and was FWE-corrected at the peak level; all clusters greater than or equal to 10 are presented. ^†^ Cluster less than 10. ^‡^ A small volume correction (SVC) was applied. MTG = middle temporal gyrus; MiddleFG = middle frontal gyrus; IFG = inferior frontal gyrus; SFG = superior frontal gyrus; IPL = inferior parietal lobule; VTA = ventral tegmental area; SN = substantia nigra.

**Table 5 brainsci-13-00417-t005:** Simple main effects of joke type under the expectation, incongruity, and resolution–elaboration conditions.

				MNI Coordinates			
Brain Region	BA	Voxels	Side	*x*	*y*	*z*	*T* Value
(1) In the expectation stage: Jokes and nonjokes (EJ > EN)							
No significant differences							
(2) In the incongruity stage: Jokes and nonjokes (IJ > IN)							
IFG	–	10	L	−6	4	14	3.72 ^‡^
(3) In the resolution–elaboration stage: Jokes and nonjokes (RJ > RN)							
MTG	39	63	R	58	−56	10	4.73 ^‡^
MFG	9	106	L	−8	44	18	5.90
	6	6 ^†^	R	10	2	66	3.71 ^‡^
MiddleFG	8	673	L	−26	18	48	6.40
IFG	9	23	L	−52	10	30	4.71 ^‡^
	45	27	R	52	38	8	4.38 ^‡^
SFG	8	43	R	18	32	54	3.81 ^‡^
IPL	40	251	R	62	−42	30	6.18
vmPFC (MFG)	10	148	L	−8	46	16	6.09
Amygdala	–	672	L	−20	−8	−14	8.35
		40	R	20	−8	−14	6.25
Midbrain	–	236	L	−4	−18	−2	6.07
Midbrain (VTA)	–	63	L	−2	−18	−4	5.27
NAc	–	10	L	−14	6	−4	4.94

Notes: The activation threshold was set to *p* < 0.05 and was FWE-corrected at the peak level; all clusters greater than or equal to 10 are presented. ^†^ Cluster less than 10. ^‡^ A small volume correction (SVC) was applied. The nonjoke conditions (EN, IN, and RN) were used as baselines. MTG = middle temporal gyrus; MFG = medial frontal gyrus; MiddleFG = middle frontal gyrus; IFG = inferior frontal gyrus; SFG = superior frontal gyrus; IPL = inferior parietal lobule; VTA = ventral tegmental area; NAc = nucleus accumbens.

**Table 6 brainsci-13-00417-t006:** Simple main effects of the three stages under the joke condition.

				MNI Coordinates			
Brain Region	BA	Voxels	Side	*x*	*y*	*z*	*T* Value
(1) Expectation > incongruity (EJ > IJ)							
MiddleFG	10	1148	R	44	48	14	8.78
	9	48	L	−24	38	42	4.12 ^‡^
IFG	–	29	R	42	46	4	8.42
	47	62	L	−32	34	−16	5.91
SFG	10	771	R	30	56	20	6.91
vmPFC (MFG)	10	204	L	−8	52	4	7.91
(2) Expectation > resolution–elaboration (EJ > RJ)							
MFG	6	346	R	6	−24	52	6.36
MiddleFG	6	54	R	20	−12	64	4.52 ^‡^
	11	54	L	−26	34	−12	4.27 ^‡^
IFG	47	28	L	−24	34	−10	4.23 ^‡^
SFG	8	14	R	28	44	42	3.70 ^‡^
vmPFC (MFG)	10	627	R	8	44	−8	7.29
Insula	13	91	L	−50	−38	18	7.73
		254	R	48	−26	22	5.69
(3) Incongruity > expectation (IJ > EJ)							
MTG	38	131	R	50	10	−22	14.00
		110	L	−50	12	−26	11.47
MFG	9	304	R	6	48	34	9.37
		479	L	−4	46	38	8.66
MiddleFG	47	112	L	−48	36	−4	7.94
	11	8 ^†^	R	44	34	−12	5.11
IFG	9	56	L	−52	20	26	7.67
		29	R	54	20	28	6.02
SFG	9	84	R	6	48	36	9.08
	8	33	L	−36	14	54	4.20 ^‡^
IPL	40	195	R	54	−48	22	9.21
Midbrain	–	12	R	20	−26	−4	4.56 ^‡^
		62	L	−6	−26	−22	4.32 ^‡^
Insula	13	49	R	48	−42	18	9.03
		13	L	−28	20	−4	5.79
(4) Incongruity > resolution–elaboration (IJ > RJ)							
IFG	9	16	R	50	−2	26	5.32
SFG	–	33	R	22	−12	68	5.87
Insula	13	832	R	36	−16	20	7.96
		106	L	−50	−36	18	6.88
(5) Resolution–elaboration > expectation (RJ > EJ)							
MTG	38	187	L	−50	12	−26	13.88
		164	R	50	10	−22	13.56
MFG	8	2670	L	−4	20	48	17.46
IFG	10	52	L	−42	40	14	5.35
SFG	9	243	R	8	54	36	9.23
	10	67	L	−22	52	26	8.28
vmPFC (MFG)	9	273	L	−6	46	18	8.19
Amygdala	–	122	L	−20	−8	−12	9.52
		25	R	20	−6	−14	5.12
Midbrain	–	1411	L	−6	−20	−2	11.41
Midbrain (SN)	–	23	L	−8	−14	−10	10.18
		27	R	8	−12	−10	9.30
NAc	–	292	R	12	10	0	12.06
		323	L	−12	12	4	11.68
Insula	13	299	R	34	24	2	13.93
		49	L	−46	−20	18	5.57
(6) Resolution–elaboration > incongruity (RJ > IJ)							
MTG	21	784	R	64	−38	−10	10.56
MFG	–	2885	L	−4	26	40	18.55
MiddleFG	9	255	R	46	32	36	9.55
	10	55	L	−38	50	14	8.97
IFG	47	958	L	−42	16	−6	13.85
	10	103	R	46	48	0	8.16
SFG	–	5036	–	0	18	54	14.15
	9	72	L	−18	48	36	8.23
IPL	40	1613	L	−50	−52	46	15.55
		1354	R	44	−50	46	12.07
vmPFC (MFG)	10	154	L	−4	46	14	8.96
		34	R	8	52	4	4.15 ^‡^
Amygdala	–	122	L	−20	−8	−12	11.26
		81	R	20	−6	−14	7.46
Midbrain	–	1319	L	−4	−14	−10	10.89
Midbrain (SN)	–	27	R	8	−12	−10	9.81
Insula	47	2667	L	−34	18	−2	17.08
	13	490	R	34	22	2	15.00

Notes: The activation threshold was set to *p* < 0.05 and was FWE-corrected at the peak level; all clusters greater than or equal to 10 are presented. ^†^ Cluster less than 10. ^‡^ A small volume correction (SVC) was applied. The nonjoke conditions (EN, IN, and RN) were used as baselines. MTG = middle temporal gyrus; MFG = medial frontal gyrus; MiddleFG = middle frontal gyrus; IFG = inferior frontal gyrus; SFG = superior frontal gyrus; IPL = inferior parietal lobule; SN = substantia nigra; NAc = nucleus accumbens.

## Data Availability

Data are available on request due to restrictions, e.g., privacy or ethical. The data presented in this study are available on request from the corresponding author. The data are not publicly available as there are still data to be analyzed and published.

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
