# Peer review of "Verification of the Four-Stage Model of Humor Processing: Evidence from an fMRI Study by Three-Element Verbal Jokes"

_brainsci, 2023, doi:10.3390/brainsci13030417_

Round 1
Reviewer 1 Report
This article is very interesting, excellent and nothing to add but congratulate the authors. Maybe once small issue is that the title refers to jokes and this could be too general. Since the study focused on verbal jokes maybe adding this aspect to the title could be clarifying but it is just a comment.
Reviewer 2 Report
This is an interesting manuscript on the dissociation of three-stage verbal jokes from fMRI.
The manuscript has many strengths so I am in favour of its publication. Among them, I would like to highlight the theoretical introduction, where the state of the art is correctly described. It also provides a large number of details that favour replication in the methodological part. Finally, I am grateful for the discussion figure that helps to visualise the direction of the discussions.
However, I would like to highlight two minor points, which I hope may be of interest to the authors.
Firstly, to check whether there are any studies exactly similar to the hypothesis put forward, and if not, to explain this to the authors' knowledge.
Secondly, would it be possible to attach the materials used?
Finally, to emphasise the theoretical and practical contributions of these results in the discussion. Thank you for the opportunity to review such interesting piece of research.
Reviewer 3 Report
In this manuscript, the authors investigated the neuroimaging substrate of the humor processing using a so called three stage verbal jokes task.
1), the abstract is written in an extremely confusing way. in the 1st sentence, the authors mentioned 4 process of humor, setup, incongruity, resolution, and appreciation; in the 2nd sentence, they mentioned "dissociation", is this a process of humor or not? in the 3rd sentence and forthcoming, they mentioned that they used a specific 3-stage structure, however, they listed 4 stages, setup expectation stage, incongruity stage, resolution stage, and appreciation stage, and how are they stages relevant to "the comprehension and appreciation processes"? is setup expectation the same as setup or not? these unclear usage of terms make it hard for the readers to understand what the authors are talking about, nor can people understand the novelty and contribution of the study.
2), relevant to the above comment, can the authors add a figure in the manuscript to explain the humor processes proposed in the field? perhaps including the various theories such as the incongruity-resolution theory, comprehension elaboration theory, and the tri-component theory, and also noting the novelty of the current study in the same figure. and can they in the figure legends or use a box to clearly define the key terminologies?
3), introduction, can the authors explain what is human comprehension and what is human appreciation? this will be more reader friendly.
are the processes of humor comprehension the same as those of humor appreciation?
can the authors elaborate a bit on the incongruity resolution theory and comprehension elaboration theory to let the readers understand the background of the current study.
4), methods, why do the authors think 54 subjects are sufficient for the current study?
is it possible that the authors can attach the jokes and nonjokes used in the study as supplementary material? this will greatly increase the impact and citation of the study. however, if the authors have copyright issues, they can choose not to attach the materials.
Round 2
Reviewer 3 Report
Thank the authors for addressing my concerns. The manuscript has been greatly enhanced.